# Does Foreign Direct Investment Successfully Lead to Sustainable Development in Singapore?

**Abdul Rahim Ridzuan [1,*], Nor Asmat Ismail [2] and Abdul Fatah Che Hamat [2]**

[1]   Faculty of Business and Management, Universiti Teknologi MARA, Shah Alam 40450, Malaysia
[2]   School of Social Sciences, Universiti Sains Malaysia, Pulau Pinang 11800, Malaysia;
      norasmat@usm.my (N.A.I.); abdfatah@usm.my (A.F.C.H.)
*   Correspondence: rahim670@staf.uitm.edu.my; Tel.: +06-016-2325105

**Abstract:** The role of foreign direct investment (FDI) inflows is tested on three main pillars of sustainable development (SD), which consists of economic growth, income distribution and environmental quality for Singapore. The analysis is performed by using Autoregressive Distributed Lag (ARDL) estimation technique. The sample data is based on annual data, covering the period from 1970 to 2013. The estimated long-run elasticity indicated that FDI inflows not only lead to higher economic growth and better environmental quality but also widen the income disparity in this country, which may disrupt its SD mission. The other two introduced variables that could also play a part as potential drivers for sustainable development (SD) are trade openness (TO) and financial development (FD). Based on the outcomes, TO has also led to higher economic growth and lower environmental degradation. However, this variable does not have significant impact on income distribution for Singapore. As for FD, it is found to have a significant and positive impact on economic growth and also successfully reduce the income inequality problem. On the contrary, this variable does not have any significant relationship with environmental quality, as indicated by carbon dioxide ($CO_2$) emissions. Mixed evidence of a relationship is detected for other macroeconomic variables in the three estimates models. As the income inequality issue has become more serious, it is important for Singaporean policymakers to focus on attracting more foreign investors to invest in various sectors, in the hope that these companies can offer better wages to the local workers and thus improve income distribution in the country. More attention is needed to explore the potential role of TO and FD as drivers for SD in this country.

**Keywords:** sustainable development; foreign direct investment; trade openness; financial development

**JEL Classification:** F18; F62; F63

## 1. Introduction

The World Conference on Environment and Development (1987) has defined sustainable development (SD) as development that meets present needs without comprising the ability of future generations to meet their own needs. This concept mainly consists of three dimensions, such as economics, social and environmental. The United Nations Conference on Trade and Development, UNCTAD (2014) has promoted foreign direct investment (FDI) as a potential driver for SD. SD goals have become an important mission, targeting many countries around the world, including Singapore. As one of the founders of the Association of South East Asian Nations (ASEAN) group, Singapore could act as a role model for other member countries in pursuing this ultimate goal. As a small island country, Singapore has done remarkably well in the decades of the twentieth century and has been one of the fastest growing economies in East Asia. Overall, the country has successfully achieved high

economic growth through free trade and investment strategies implementation, which has led the country to become the top destination for investment, due to its favorable lending rates to foreign investors, implementation of a simple regulatory system, availability of tax incentives, a high-quality infrastructure, political stability, strong financial market and the absence of corruption. In the recent world investment report by UNCTAD (2015), Singapore is the fifth largest recipient of FDI in the world and the third largest of the East and Southeast Asian countries. Based on Figure 1 below, the country experienced a drastic fall in FDI inflows in 2007 as a result of the global economic crisis. The fall in 2008 was basically far greater than in late 1997, which happened during the Asian financial crisis. Interestingly, one year after the event, the FDI inflows into Singapore drastically increased and achieved its highest point in 2013, with a total value of US$64,793,170,000. Overall, the success of this country in attracting a higher inflow of FDI is due to the effectiveness of its Economic Development Board (EDB) that maintains a network of overseas promotion offices and an international advisory council that includes global heads of leading multinational corporations (MNC).

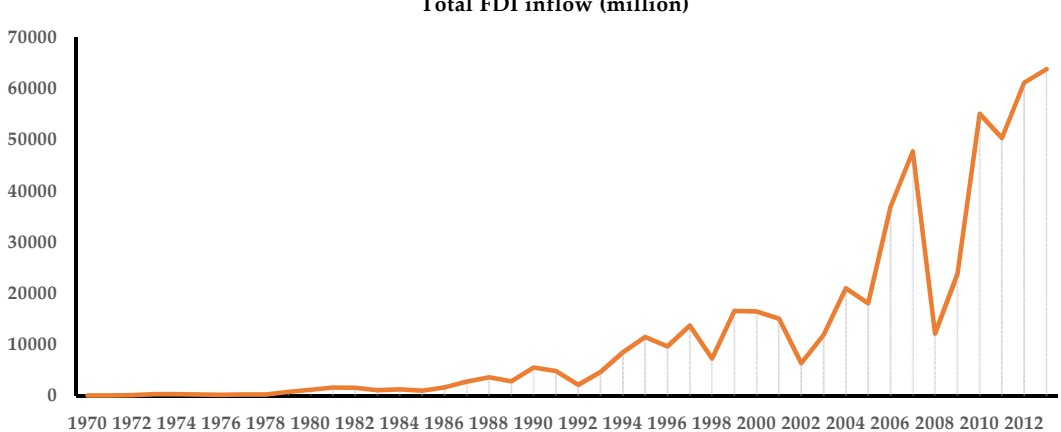

**Figure 1.** The trend of total FDI inflows in Singapore from 1970 to 2013.

Apart from achieving high economic growth, the country is also very concerned about the risks of negative externalities, such as pollution, as a result of industrialization in the country. To keep pollution under control, the Singapore government has implemented very strict regulatory measures. The third pillar for SD, known as income distribution, proxied by the GINI coefficient, was also observed in this study. Besides increasing the trend of $CO_2$ emissions, there is also an increasing trend of income inequality in the country. Singapore's income gap is one of the widest among developed countries. To narrow this gap, the government has made some effort to raise wages at the bottom and increase taxes on wealth at the top. However, relying only on the government side to maintain environmental quality as well as narrowing the income gap may not be enough in the long run. Thus, this research paper would like to test the potential role of FDI inflows that could bring favorable impacts on economic growth, income distribution and environmental quality, which represent the three pillars of SD. Given a strong record of FDI inflows to Singapore, this variable could be the best indicator for the realization of SD as suggested by UNCTAD. The findings of this research paper could help the country to prepare a better road map for achieving SD as this goal is also being targeted under ASEAN Vision 2025 as well as Sustainable Development Goal (SDG) 2030. The rest of the paper is structured as follows: Section 2 briefly explains the literature review; Section 3 focuses on the methodology; Section 4 displays data; Section 5 presents the empirical analysis and the last section concludes the paper with policy recommendations.

## 2. Literature Review

In this section, the empirical studies between FDI inflows and the three pillars of SD were examined. Various determinants, techniques of analysis, sample data and country selection have been used in this area of studies. The findings from previous studies would help to give better insights on the potential role of FDI inflows towards economic growth, income distribution and environmental quality.

### 2.1. Foreign Direct Investment and Economic Growth

The literature on the question of whether FDI contributes to economic growth was inspired by both advances in the endogenous growth literature and by a reasonable desire to provide guidance for policymakers. Duttaray et al. (2008) used data from 66 developing countries to research the causality between FDI and economic growth. FDI was found to affect growth in 29 of these countries, but growth was not found to affect FDI at all. Wang and Wong (2009) used data from 12 Asian countries from 1987 to 1997 to find out whether FDI has an influence on economic growth. Even though endogenous growth theory predicts a positive relationship between inward FDI and economic growth, Wang and Wong (2009) suggested that using total FDI could blur its effects and lead to ambiguous results. The study used FDI in different sectors and concluded that FDI enhanced economic growth in manufacturing but not in non-manufacturing sectors. Gui-Diby (2014) examined the impact of FDI on economic growth in 50 African countries during the period from 1980 to 2009 by using system Generalized method-of-moments (GMM) estimator as proposed by Blundell and Bond (1998). The research revealed that FDI inflows had a significant impact on economic growth in the African region during the period of interest. Studies by Pegkas (2015) found a long run positive relationship between FDI stock and economic growth in Eurozone countries during the period of 2002–2012. The author employed panel data estimations and ran the regression using Full Modified Ordinary Least Square (FMOLS) and Dynamic Ordinary Least Square (DOLS), which have properties to remove endogeneity problem in the econometric model. The elasticity of GDP with respect to FDI was 0.054% and 0.147%, respectively. As FDI plays a significant role in economic growth, the author recommended macroeconomic stability and reduction of market distortion, which both are necessary for the creation of a suitable environment to attract FDI. Sunde (2017) enriched the empirical literature review on this scope of study by investigating the impact of export and FDI inflows towards economic growth in South Africa. The author adopted Bound estimation and the sample period of the study was taken from 1990 to 2014. The main findings obtained from this research showed that both FDI inflows and exports lead towards positive economic growth in South Africa thus confirming FDI led growth and export led growth hypothesis. The author proceeds with VECM granger causality test to add more insights for policymakers and found unidirectional causality between economic growth and FDI inflows running from FDI inflows to economic growth, unidirectional causality between FDI inflows and export running from FDI inflows to exports and bidirectional causality between economic growth and export.

### 2.2. Foreign Direct Investment and Income Distribution

Earlier studies, such as Basu and Guariglia (2007), used a panel of around 80 countries to test a theoretical model linking FDI to growth and inequality in human capital and concluded that inward FDI promoted economic inequality. Other studies, such as by Sylwester (2005) and Adams (2009) found that FDI variables were not statistically significant in explaining income inequalities. In country-specific studies by Zhang and Zhang (2003), it was revealed that increasing FDI inflows contributed to greater income inequality in China. Wei et al. (2009) blamed the uneven distribution of FDI (rather than FDI itself) to be the cause of rising regional inequality in China. Jensen and Guillermo (2007) revealed a decrease in income inequality in Mexico with FDI inflows whereas Mah (2003) found no significant effect for Korea. Blonigen and Slaughter (2001) failed to find any significant effects of FDI on wage inequality between skilled and unskilled workers in the United States. In contrast,

Chintrakarn et al. (2012) found that FDI exerted a significant and robust negative effect on income inequality in the United States, but with much heterogeneity across states. The inconclusive empirical evidence may be suggestive of nonlinearity in the link between FDI and inequality. Figini and Görg (2011) revealed that for developing countries, wage inequality increases with FDI, but this effect diminishes with further increases in FDI. In contrast, wage inequality decreases with FDI for developed countries. Lin et al. (2013) found that FDI increases inequality in income distribution when a country achieves a threshold of human capital between 6.0 and 6.7 years of secondary schooling. Below this threshold, however, FDI improves income distribution. In this article, the authors considered financial development as a potential vehicle in shaping nonlinearity in the link between FDI and inequality. Hyungsun and Miguel (2017) in their research paper aims to investigate the relationship between FDI inflows and FDI stock on income distribution for 7 Southeast Asia countries consists of Cambodia, Indonesia, Laos, Malaysia, Philippines, Thailand, and Vietnam. The study was conducted based on sample period range from 1990 to 2013. Besides FDI inflows and FDI stock, the authors introduce other control variables. The main outcome generated from panel FMOLS revealed that higher FDI inflows will worsen income equality in this group of countries. The square term of FDI stock, on the other hand, found to be noteworthy with according to the expected sign. The findings of other control variable such as trade and GDP per capita were not significant.

## 2.3. Foreign Direct Investment and Environmental Quality

Based on the model of environmental quality, Cole and Elliott (2005) found that capital flows from Japan to Southeast Asian countries is likely to increase the $CO_2$ emissions level in the recipient countries because most of the investments are focused on heavy and polluting industries. According to Cole et al. (2006), the less developed countries are always the best choice for investment due to the low stringent environmental policies. However, in China, Wu and Li (2011) concluded that the increase in FDI does not increase the level of pollution. They added that foreign-owned industries are not the source of pollution because they have more reasonable industrial structure and advanced technology than domestic industries. Therefore, foreign-owned industries will encourage technology transfer, i.e. efficiency in resource utilization and thus reduce the level of pollution. Furthermore, Kirkulak et al. (2011) and List et al. (2004) also found the same results in China and the United States. Nonetheless, He (2006) explored the pollution haven hypothesis (PHH) and the environmental impact of FDI in 29 Chinese industrial provinces, using the generalized method of moment (GMM) estimator. The study arrived at the conclusion that FDI has a small positive effect on pollution via its impact on output grow within the Chinese provinces. Cole et al. (2011) employed the panel data approach to examine the impact of FDI and output on environmental quality in 112 major cities in China. They found that FDI and output have a significant positive effect on air and water pollution in these major cities. Therefore, these studies showed some supporting evidence for the pollution haven hypotheses in the Chinese economy. In the recent research by Behera and Dash (2017), the authors examines the relationship between urbanization, energy consumption, FDI and $CO_2$ emissions of 17 countries (categorized into three sub-group consist of high, middle and low-income group) in the South and Southeast Asian (SSEA) region based on period 1980 until 2012. Based on Pedroni cointegration tests, the evidence of long run relationship is detected between the tested variables. However, when the authors switched to Westerland cointegration test, the long run relationship only exists for middle-income countries. Overall results showed that energy consumption have a stronger negative influenced towards environmental quality as compared to FDI inflows in both high and middle-income countries whereas there is no substantial effect of FDI and urbanization for low-income group. The huge consumption of fossil fuel type of energy by middle-income group could hinder these countries to achieve SD goal. As for policy recommendation, there is an urgency for these countries to embrace more energy conservation polices to reduce the emissions. As both urbanization and FDI inflows contribute towards environmental degradation, the authors also suggest that the countries need to adopt sustainable urbanization model and pollution induced growth model.

## 3. Methodology

The three proposed econometric models are introduced in this section. All variables are transformed into log-linear form (LN) and thus the estimated results from these models represent elasticities. According to Shahbaz et al. (2013), modeling the log-log model specification will provide efficient results by reducing the sharpness in time series data compared with the simple linear-linear specification.

### 3.1. Model of Economic Growth

A standard augmented Cobb-Douglas production function framework with FDI as an additional variable along with capital and labor was introduced as in the following equation:

$$Y_t = f\ (K, L, FDI) \tag{1}$$

where *Y* is real output, *K* is capital, *L* is labor and *FDI* is foreign direct investment inflows. Following Grossman and Elhanan (1991) and Barro and Sala-I-Martin (1995), this production's function has been extended according to the new growth theory also known as endogenous growth model introduced by Lucas (1988), Romer (1986), and Rebelo (1991). The endogenous growth theories show that the long run growth of a country is not only inclined by the volume of physical investment, but also hang on the efficiency of utilizing investment. Equation (1) above is log and extended by including other important variables as shown in Equation (2) below

$$LNGDP_t = \beta_0 + \beta_1 LNLAB_t + \beta_2 LNDI_t + \beta_3 LNFDI_t + \beta_4 LNHC_t + \beta_5 LNTO_t + \beta_6 LNFD_t + \varepsilon_t \tag{2}$$

where *GDP* is real gross domestic product per capita constant 2005; *LAB* is total labor force; *DI* is gross fixed capital formation constant 2005; *FDI* is foreign direct investment inflows as a percentage of *GDP*; *HC* is secondary school enrolment rate; *TO* is the sum of export and import over *GDP* and *FD* is financial development proxy by money supplied, M2 over *GDP*. *LAB* is introduced in the model to control the additional determinant of growth, *GDP* in order to reduce the problem of omitted variable bias. Domestic investment, *DI* is included following an extended model proposed by Barro (1999). Next, *FDI* highlighted by Sahoo and Mathiyazhagan (2003) is added as it can potentially bring a positive impact on Singapore economic growth. Meanwhile, *HC* which is based on Barro and Lee (2011) database is added as it plays a critical role in absorbing foreign knowledge or skills brought in by a foreign investor from FDI inflows. Trade openness (TO) is expected to have a positive effect as stated by Balasubramanyam et al. (1996). We also added financial development (FD) to act as another control variable and it is also expected to have a positive impact on growth especially for advanced country like Singapore. The unrestricted error correction model (UECM) form of the ARDL model that contains both the short run and long run dynamics for a model of economic growth is shown as follows:

$$\begin{aligned}
\Delta LNGDP_t &= \beta_0 + \lambda_0 LNGDP_{t-1} + \lambda_1 LNLAB_{t-1} + \lambda_2 LNDI_{t-1} + \lambda_3 LNFDI_{t-1} + \lambda_4 LNHC_{t-1} \\
&+ \lambda_5 LNTO_{t-1} + \lambda_6 LNFD_{t-1} + \sum_{i=1}^{p} \omega_i \Delta LNGDP_{t-i} + \sum_{i=0}^{q} \gamma_i \Delta LNLAB_{t-i} + \sum_{i=0}^{r} \delta_i \Delta LNDI_{t-i} \\
&+ \sum_{i=0}^{s} \pi_i \Delta LNFDI_{t-i} + \sum_{i=0}^{t} \vartheta_i \Delta LNHC_{t-i} + \sum_{i=0}^{u} \zeta_i \Delta LNTO_{t-i} + \sum_{i=0}^{v} \psi_i \Delta LNFD_{t-i} + \mu_t
\end{aligned} \tag{3}$$

where $\Delta$ is the first difference operator and $\mu_t$ is the white-noise disturbance term. Residuals for all the UECM models should be serially uncorrelated and the models should be stable. The above final model for economic growth can be viewed as an ARDL or Bound of order (*p, q, r, s, t, u, v*).

## 3.2. Model of Income Distribution

The model of income distribution introduced in this study was a modified version of the model introduced by Mah (2003). The original model proposed by Mah (2003) is listed as follows:

$$GINI_t = f\ (Y_t, Y^2{}_t, FDI_t, TO_t) \tag{4}$$

where *GINI* is Gini coefficient; *Y* and *Y* square representing per capita income and per capita income square respectively, followed by foreign direct investment inflows (*FDI*) and trade openness (*TO*). As the study not intend to validate the existence of Kuznets Curve, we remove Y square from Equation (4) above. Instead, we introduce several other crucial variables such as domestic investment (DI) and financial development (FD) which is seen as follows

$$LNGINI_t = \alpha_0 + \alpha_1 LNGDP_t + \alpha_2 LNDI_t + \alpha_3 LNFDI_t + \alpha_4 LNTO_t + \alpha_5 LNFD_t + \varepsilon_t \tag{5}$$

where *GINI* is Gini coefficient; *GDP* is real gross domestic product per capita constant 2005; *DI* is gross fixed capital formation constant 2005; *FDI* is foreign direct investment inflows as a percentage of *GDP*; *TO* is the sum of export and import over *GDP* and *FD* is financial development proxied by the ratio of money supply, M2 to *GDP*. *GDP* (gross domestic product constant 2005) is expected to have either a positive or negative impact on income distribution. The expansion of Singapore economy is expected to reduce the income disparity in the society as more job opportunities are being produced. Besides GDP, other determinants such as DI, FDI, TO, and FD are also expected to have either a positive or negative relationship with GINI. According to Mah (2003), DI is expected to have a positive relationship with GINI while the relationship is expected to be negative according to the neoclassical theorist. The negative relationship between DI and GINI is explained by the intuition that increases in domestic investment spending mean more people getting jobs, which implies that more people are earning, thereby putting a downward pressure on income inequality. Next, according to Mundell hypothesis, FDI is expected to have a negative relationship with GINI. Contrary to Mundell hypothesis, Feenstra and Hanson (1997) postulated that FDI has a positive relationship with GINI. Meanwhile, according to dependency theorist, TO is expected to have a positive association with the country's income distribution. Lastly, according to Greenwood and Jovanvoic (1990), FD initially increases income inequality but decreases income inequality once the financial sector matures and thus may posit a negative relationship in Singapore. The UECM for this model is listed as follows:

$$\Delta LNGINI_t = \beta_0 + \rho_0 LNGINI_{t-1} + \rho_1 LNGDP_{t-1} + \rho_2 LNDI_{t-1} + \rho_3 LNFDI_{t-1} + \rho_4 LNTO_{t-1}$$
$$+\rho_5 LNFD_{t-1} + \sum_{i=1}^{p} \omega_i \Delta LNGINI_{t-i} + \sum_{i=0}^{q} \gamma_i \Delta LNGDP_{t-i} + \sum_{i=0}^{r} \delta_i \Delta LNDI_{t-i} + \sum_{i=0}^{s} \pi_i \Delta LNFDI_{t-i}$$
$$+\sum_{i=0}^{t} \vartheta_i \Delta LNTO_{t-i} + \sum_{i=0}^{u} \zeta_i \Delta LNFD_{t-i} + \mu_t \tag{6}$$

where $\Delta$ is the first difference operator and $\mu_t$ is the white-noise disturbance term. The above final model for economic growth can be viewed as bound of order (*p*, *q*, *r*, *s*, *t*, *u*).

## 3.3. Model of Environmental Quality

The environmental quality model in this study was basically based on the econometric model proposed by Shahbaz et al. (2013). The general form of the empirical model is shown as follows:

$$CO_{2t} = f\ (Y_t, F_t, E_t, TR_t) \tag{7}$$

where $CO_2$ is emissions metric ton per capita, *Y* is real GDP per capital, *F* is financial development, *E* is energy consumption per capita, and *TR* is trade openness per capita. The above equation was tested on Indonesia. As for the case of Singapore, we added additional variable namely foreign direct

investment into the equation as Singapore recorded the highest amount of FDI inflows as compared to other Association of South East Asian Nation (ASEAN) member countries. Thus, there is some possibilities that FDI inflows might influence the level of environmental quality in this country. The log version of the equation can be seen as follows:

$$LNCO_{2t} = \phi_0 + \phi_1 LNGDP_t + \phi_2 LNFDI_t + \phi_3 LNFD_t + \phi_4 LNEN_t + \phi_5 LNTO_t + \varepsilon_t \tag{8}$$

where $CO_2$ is emissions metric ton per capita; $GDP$ is real gross domestic product per capita constant 2005; $FDI$ is foreign direct investment inflows as a percentage of $GDP$; $FD$ is financial development measured by money supply, M2 to $GDP$, $EN$ is per capita energy consumption, kg of oil equivalent and $TO$ is sum of export and import over $GDP$. The model as listed in Equation (3) above is a modified version of Lee (2013) model. $CO_2$ emissions is used as a proxy for environmental quality because this gas has an impact in worsening the greenhouse effect and is the main culprit for more than 60% of the greenhouse effect (Ozturk and Acaravci 2010). Besides, the main reasons for using carbon emissions is that they play a central role in the current debate on the environmental protection, climate change abatement, energy security, sustainable use of available resources and sustainable development. $GDP$ is expected to have a positive sign with the level of $CO_2$ emissions. $FDI$ is expected to have a negative sign with $CO_2$ emissions if foreign investors used clean technology in their production besides having to obligate strict environmental rules set by the Singapore government. Besides $FDI$, $TO$ may also deteriorate or improve environmental quality, Antweiler et al. (2001). Other potential determinants, such as $EN$, were added, as recommended by Ang (2008). $FD$ is also expected to have a mixed sign. In the case of Singapore, we expect that there will be a negative sign between $FD$ and $CO_2$ emissions. As the financial sector in Singapore has reached maturity, it could improve environmental quality through the allocation of some financial resources to firms to utilize environmental friendly technology. The UECM for this model is listed as follows:

$$\Delta LNCO_{2t} = \beta_0 + \kappa_0 LNCO_{2t-1} + \kappa_1 LNGDP_{t-1} + \kappa_2 LNFDI_{t-1} + \kappa_3 LNFD_{t-1} + \kappa_4 LNENY_{t-1}$$
$$+ \kappa_5 LNTO_{t-1} + \sum_{i=1}^{p} \omega_i \Delta LNCO_{2t-i} + \sum_{i=0}^{q} \gamma_i \Delta LNGDP_{t-i} + \sum_{i=0}^{r} \delta_i \Delta FDI_{t-i} + \sum_{i=0}^{s} \pi_i \Delta LNFD_{t-i} \tag{9}$$
$$+ \sum_{i=0}^{t} \vartheta_i \Delta LNENY_{t-i} + \sum_{i=0}^{u} \zeta_i \Delta LNTO_{t-i} + \mu_t$$

where $\Delta$ is the first difference operator and $\mu_t$ is the white-noise disturbance term. The above final model for economic growth can be viewed as bound of order ($p, q, r, s, t, u$). To test for cointegration among the variables for the proposed models above, we test the combine hypothesis of no cointegration on the level variables in Equations (3), (6) and (9), which is $H_0$: $\lambda_0 = \lambda_1 = \lambda_2 = \lambda_3 = \lambda_4 = \lambda_5 = \lambda_6 = 0$ for the model of economic growth (Equation (3)), $H_0$: $p_0 = p_1 = p_2 = p_3 = p_4 = p_5 = 0$ for the model of income distribution (Equation (6)) and $H_0$: $k_0 = k_1 = k_2 = k_3 = k_4 = k_5 = 0$ for the model of environmental quality (Equation (9)) against $Ha$: $\lambda_0 \neq \lambda_1 \neq \lambda_2 \neq \lambda_3 \neq \lambda_4 \neq \lambda_5 \neq \lambda_6 \neq 0$ for the model of economic growth, $Ha$: $p_0 \neq p_1 \neq p_2 \neq p_3 \neq p_4 \neq p_5 \neq 0$ for model of income distribution and $Ha$: $k_0 \neq k_1 \neq k_2 \neq k_3 \neq k_4 \neq k_5 \neq 0$ model of environmental quality, which suggests the existence of cointegration among the variables. The presence of cointegration was based on $F$-statistic value. This $F$-statistic value is then compared with critical value introduced by Pesaran et al. (2001). If the estimated $F$-statistics is greater than the upper bound value of the table, we can reject the null hypothesis and accept the alternative hypothesis that cointegration exists. On the other hand, if $F$-statistic is lower than the lower bound, the null hypothesis cannot be rejected and thus the cointegration in the proposed model does not exist. Given that the long run elasticities were unable to show us the causality between the determinants in each model, we also performed Toda-Yamamoto granger non-causality test that could provide more information for policy recommendations.

## 4. Data

This study used an annual data starting from 1970 to 2013 comprising 44 years, as a sample period. Data such as DI, EN, FD, FDI, GDP, LAB and TO were taken from World Development Indicator (WDI) 2016. HC data was taken from Barro and Lee database, LAB was completed by using data from International Labour Organisation (ILO 2015), $CO_2$ emissions data was secured using both (WDI 2016) and Emissions Database for Global Atmospheric Research (EDGAR 2015) and GINI data was sourced from the University Texas Income Project (UTIP 2008) and Global Consumption Income Project (GCIP 2015).

## 5. Empirical Findings

The first analysis performed was the descriptive statistics analysis that best described the basic information of the variables of the three econometric models. Table 1 displays the information such as mean, median, maximum, minimum, standard deviation, skewness, and kurtosis. For example, the highest mean of LNEN (12.08) detected is from Indonesia while the lowest mean (6.11) is recorded from the Philippines. This means that Indonesia used the highest amount of energy consumption due to its huge population as compared to the Philippines who used less energy consumption in the country. Meanwhile, the mean and median that implied the normal distribution of the data for every variable in each ASEAN-5 country were close enough to each, thus, provided more robust analysis. The minimum and the maximum value showed that the there was an overall increasing trend in the variables. Meanwhile, the standard deviation revealed that the average or typical distance scores varied from the mean. For example, in the case of Malaysia, the standard of typical distance that LNGDP values varied or spread from the mean was by about 0.482, whether its 0.482 above 8.126 or 0.482 below than 8.126.

**Table 1.** Descriptive Analysis.

|  | LNGDP | LNCO$_2$ | LNGINI | LNLAB | LNDI | LNFDI | LNHC | LNTO | LNEN | LNFD |
|---|---|---|---|---|---|---|---|---|---|---|
| | | | | | Malaysia | | | | | |
| Mean | 8.126 | 1.268 | 8.304 | 7.293 | 3.290 | 1.144 | 9.339 | 4.834 | 7.183 | 4.585 |
| Median | 8.147 | 1.332 | 8.298 | 8.849 | 3.231 | 1.286 | 9.354 | 4.932 | 7.286 | 4.767 |
| Maximum | 8.853 | 2.080 | 8.486 | 9.500 | 3.774 | 2.170 | 10.200 | 5.333 | 7.891 | 4.968 |
| Minimum | 7.232 | 0.291 | 8.098 | 3.608 | 2.900 | −2.870 | 8.243 | 4.231 | 6.261 | 3.609 |
| Std Dev. | 0.482 | 0.598 | 0.070 | 2.512 | 0.229 | 0.794 | 0.491 | 0.388 | 0.545 | 0.379 |
| Skewness | −0.203 | −0.093 | 0.069 | −0.653 | 0.715 | −3.112 | −0.329 | −0.168 | −0.254 | −1.092 |
| Kurtosis | 1.799 | 1.475 | 3.752 | 1.460 | 2.393 | 16.195 | 2.517 | 1.355 | 1.731 | 3.066 |
| | | | | | Indonesia | | | | | |
| Mean | 6.718 | −0.080 | 8.299 | 11.221 | 3.167 | 0.610 | 9.091 | 4.084 | 12.088 | 3.380 |
| Median | 6.828 | 0.011 | 8.296 | 11.254 | 3.201 | 0.657 | 9.071 | 4.071 | 12.129 | 3.645 |
| Maximum | 7.501 | 0.667 | 8.592 | 11.720 | 3.486 | 1.613 | 10.350 | 4.318 | 12.920 | 4.092 |
| Minimum | 5.806 | −1.158 | 7.972 | 10.663 | 2.771 | −1.323 | 7.042 | 3.833 | 11.180 | 2.209 |
| Std Dev. | 0.486 | 0.525 | 0.186 | 0.323 | 0.188 | 0.648 | 0.811 | 0.143 | 0.489 | 0.534 |
| Skewness | −0.272 | −0.378 | −0.393 | −0.118 | −0.245 | −1.030 | −0.552 | −0.117 | 0.004 | −0.497 |
| Kurtosis | 1.902 | 2.122 | 1.831 | 1.835 | 2.332 | 4.139 | 3.175 | 1.803 | 1.963 | 1.832 |
| | | | | | Thailand | | | | | |
| Mean | 7.321 | 0.515 | 8.408 | 9.685 | 3.320 | 0.346 | 9.310 | 4.534 | 6.668 | 4.282 |
| Median | 7.466 | 0.650 | 8.382 | 10.361 | 3.279 | 0.340 | 9.422 | 4.625 | 6.707 | 4.383 |
| Maximum | 8.142 | 1.520 | 8.544 | 11.000 | 3.728 | 1.877 | 10.160 | 5.097 | 7.500 | 4.901 |
| Minimum | 6.369 | −0.874 | 8.294 | 7.406 | 3.036 | −1.599 | 8.217 | 3.989 | 5.800 | 3.467 |
| Std Dev. | 0.577 | 0.781 | 0.073 | 1.247 | 0.197 | 0.884 | 0.599 | 0.397 | 0.570 | 0.457 |
| Skewness | −0.230 | −0.174 | 0.351 | −1.105 | 0.883 | −0.217 | −0.390 | −0.035 | 0.041 | −0.322 |
| Kurtosis | 1.604 | 1.461 | 1.792 | 2.322 | 2.689 | 2.088 | 1.938 | 1.363 | 1.445 | 1.617 |

**Table 1.** *Cont.*

|  | LNGDP | LNCO$_2$ | LNGINI | LNLAB | LNDI | LNFDI | LNHC | LNTO | LNEN | LNFD |
|---|---|---|---|---|---|---|---|---|---|---|
| | | | | | Philippines | | | | | |
| Mean | 6.972 | −0.253 | 8.413 | 9.325 | 3.059 | 0.637 | 9.039 | 4.174 | 6.116 | 3.644 |
| Median | 6.941 | −0.214 | 8.423 | 10.057 | 3.030 | 0.647 | 9.044 | 4.212 | 6.111 | 3.571 |
| Maximum | 7.366 | −0.022 | 8.486 | 10.615 | 3.396 | 1.427 | 9.810 | 4.690 | 6.241 | 4.244 |
| Minimum | 6.716 | −0.661 | 8.309 | 7.166 | 2.775 | −0.282 | 8.240 | 3.607 | 6.000 | 2.973 |
| Std Dev. | 0.152 | 0.154 | 0.041 | 1.370 | 0.151 | 0.419 | 0.457 | 0.382 | 0.058 | 0.416 |
| Skewness | 0.791 | −0.951 | −0.581 | −0.729 | 0.180 | −0.126 | −0.070 | −0.219 | 0.226 | −0.082 |
| Kurtosis | 3.128 | 3.370 | 2.684 | 1.629 | 2.353 | 2.206 | 1.819 | 1.416 | 2.606 | 1.360 |
| | | | | | Singapore | | | | | |
| Mean | 9.683 | 2.267 | 8.309 | 7.301 | 3.488 | 2.301 | 8.960 | 5.493 | 8.152 | 4.440 |
| Median | 9.769 | 2.490 | 8.265 | 7.350 | 3.506 | 2.318 | 9.092 | 5.509 | 8.379 | 4.421 |
| Maximum | 10.515 | 2.950 | 8.480 | 7.990 | 3.533 | 3.277 | 10.000 | 6.108 | 8.909 | 4.890 |
| Minimum | 8.488 | 0.800 | 8.171 | 6.546 | 3.138 | 1.313 | 7.758 | 4.382 | 7.130 | 3.989 |
| Std Dev. | 0.596 | 0.558 | 0.103 | 0.425 | 0.181 | 0.543 | 0.628 | 0.467 | 0.539 | 0.280 |
| Skewness | −0.349 | −1.360 | 0.448 | −0.127 | −0.158 | −0.066 | −0.350 | −0.564 | −0.301 | 0.129 |
| Kurtosis | 1.922 | 3.867 | 1.668 | 1.967 | 2.458 | 2.034 | 2.034 | 2.449 | 1.663 | 1.647 |

Note: Std. Dev is standard deviation.

The second analysis began with testing the stationarity of the data. This step was important to make sure that variables used in the study were not integrated of order more than I (1). The presence of I (2) and beyond could violate the requirements of using the ARDL estimation and critical value table proposed by Pesaran et al. (2001) and Narayan and Narayan (2005). Based on the results of Augmented Dickey-Fuller (ADF) and Philipp Perron (PP) displayed in Table 2 below, it can be seen that there is a mix stationarity either at I (0) or I (1) (at the level or at first difference) and thus justifies the use of ARDL cointegration test.

**Table 2.** Results of ADF and PP Unit Root test.

| Model | | Variable | ADF Test Statistic | | PP Test Statistic | |
|---|---|---|---|---|---|---|
| | | | Intercept | Trend and Intercept | Intercept | Trend and Intercept |
| Model of Economic Growth | Level | LNGDP | −2.97 (0) ** | −1.94 (0) | −5.53 (9) *** | −1.86 (4) |
| | | LNLAB | −0.81 (0) | −2.37 (0) | −0.81 (3) | −2.50 (1) |
| | | LNDI | −1.70 (1) | −2.72 (1) | −1.38 (2) | −2.62 (2) |
| | | LNFDI | −3.10 (0) ** | −5.31 (4) *** | −3.10 (0) ** | −6.76 (14) *** |
| | | LNHC | −1.26 (0) | −1.78 (0) | −1.19 (1) | −1.78 (0) |
| | | LNTO | −3.78 (0) *** | −2.55 (0) | −3.52 (3) ** | −2.58 (3) |
| | | LNFD | −0.30 (0) | −4.02 (0) ** | −0.08 (3) | −3.97 (3) ** |
| | First difference | LNGDP | −5.64 (1) *** | −6.67 (1) *** | −5.34 (2) *** | −7.91 (10) *** |
| | | LNLAB | −5.76 (0) *** | −5.75 (0) *** | −5.75 (2) *** | −5.72 (3) *** |
| | | LNDI | −4.56 (0) *** | −4.49 (0) *** | −4.56 (3) *** | −4.50 (3) *** |
| | | LNFDI | −6.76 (4) *** | −6.66 (4) *** | −26.58 (41) *** | −26.11 (41) *** |
| | | LNHC | −4.89 (0) *** | −4.90 (0) *** | −4.84 (3) *** | −4.83 (4) *** |
| | | LNTO | −4.81 (0) *** | −5.46 (0) *** | −4.78 (3) *** | −5.39 (5) *** |
| | | LNFD | −6.97 (0) *** | −6.90 (0) *** | −8.11 (7) *** | −8.08 (7) *** |
| Model of Income Distribution | Level | LNGINI | −0.98 (0) | −1.96 (0) | −1.30 (4) | −1.96 (2) |
| | | LNGDP | −2.97 (0) ** | −1.94 (0) | −5.53 (9) *** | −1.86 (4) |
| | | LNDI | −1.70 (1) | −2.72 (1) | −1.38 (2) | −2.62 (2) |
| | | LNFDI | −3.10 (0) ** | −5.31 (4) *** | −3.10 (0) ** | −6.76 (14) *** |
| | | LNTO | −3.78 (0) *** | −2.55 (0) | −3.52 (3) ** | −2.58 (3) |
| | | LNFD | −0.30 (0) | −4.02 (0) ** | −0.08 (3) | −3.97 (3) ** |
| | First difference | LNGINI | −5.60 (0) *** | −6.22 (0) *** | −5.70 (3) *** | −6.26 (2) *** |
| | | LNGDP | −5.64 (1) *** | −6.67 (1) *** | −5.34 (2) *** | −7.91 (10) *** |
| | | LNDI | −4.56 (0) *** | −4.49 (0) *** | −4.56 (3) *** | −4.50 (3) *** |
| | | LNFDI | −6.76 (4) *** | −6.66 (4) *** | −26.58 (41) *** | −26.11 (41) *** |
| | | LNTO | −4.81 (0) *** | −5.46 (0) *** | −4.78 (3) *** | −5.39 (5) *** |
| | | LNFD | −6.97 (0) *** | −6.90 (0) *** | −8.11 (7) *** | −8.08 (7) *** |

**Table 2.** *Cont.*

| Model | Variable | ADF Test Statistic | | PP Test Statistic | |
|---|---|---|---|---|---|
| | | Intercept | Trend and Intercept | Intercept | Trend and Intercept |
| Model of Environmental Quality | Level | | | | |
| | $LNCO_2$ | 2.29 (2) | 1.43 (2) | 1.19 (3) | −0.30 (1) |
| | LNGDP | −2.97 (0) ** | −1.94 (0) | −5.53 (9) *** | −1.86 (4) |
| | LNFDI | −3.10 (0) ** | −5.31 (4) *** | −3.10 (0) ** | −6.76 (14) *** |
| | LNFD | −0.30 (0) | −4.02 (0)** | −0.08 (3) | −3.97 (3) ** |
| | LNEN | −1.51(0) | −2.19(0) | −1.52(2) | −2.23(1) |
| | LNTO | −3.78 (0) *** | −2.55 (0) | −3.52 (3) ** | −2.58 (3) |
| | First difference | | | | |
| | $LNCO_2$ | −0.63 (4) | −8.59 (1) *** | −6.16 (3) *** | −7.95 (3) *** |
| | LNGDP | −5.64 (1) *** | −6.67 (1) *** | −5.34 (2) *** | −7.91 (10) *** |
| | LNFDI | −6.76 (4) *** | −6.66 (4) *** | −26.58 (41) *** | −26.11 (41) *** |
| | LNFD | −6.97 (0) *** | −6.90 (0) *** | −8.11 (7) *** | −8.08 (7) *** |
| | LNEN | −6.55(0) *** | −6.50(0) *** | −6.58(2) *** | −6.54(3) *** |
| | LNTO | −4.81 (0) *** | −5.46 (0) *** | −4.78 (3) *** | −5.39 (5) *** |

Note: 1. ***, ** and * are 1%, 5% and 10% of significant levels, respectively. 2. The optimal lag length is selected automatically using the Akaike Information Criteria (AIC) for ADF test and the bandwidth had been selected by using the Newey–West method for PP unit root test.

To test the existence of cointegration relationship among the variables, we performed ARDL bounds *F*-test for cointegration that was displayed in Table 3 below. The maximum lag of 4 was set in each proposed model using Akaike Information criterion (AIC). The value of *F*-statistics for each model (5.54, 3.25 and 3.97) was higher than the upper I (1) critical value table (for *k* = 6 and 5) and significant at 1, 10 and 5 per cent level, respectively. Thus, this confirmed the existence of a long-run relationship between all variables in the three models

**Table 3.** Results of ARDL bounds *F*-test for cointegration.

| Model | Max. Lag | Lag Order | F Statistic |
|---|---|---|---|
| Model of Economic Growth | 4 | (1,1,1,3,1,2,3) | 5.54 *** |
| Model of Income Distribution | 4 | (4,4,4,4,4,4) | 3.25 * |
| Model of Environmental Quality | 4 | (3,4,4,4,3,4) | 3.97 ** |

| | *k* = 6 | | *k* = 5 | |
|---|---|---|---|---|
| Critical Values for *F*-statistics [#] | Lower I (0) | Upper I (1) | Lower I (0) | Upper I (1) |
| 1% | 3.15 | 4.43 | 3.41 | 4.68 |
| 5% | 2.45 | 3.61 | 2.62 | 3.79 |
| 10% | 2.12 | 3.23 | 2.26 | 3.35 |

Note: 1. # The critical values are based on Pesaran et al. (2001), case III: unrestricted intercept and no trend. 2. *k* is a number of variables 3. *, **, and *** represent 10%, 5% and 1% level of significance, respectively. 4. *k* = 5 for a model of income distribution and model of environmental quality while *k* =6 for a model of economic growth.

Next, before we proceed with long run and short run elasticities outcome, it is very important to make sure that our proposed models are reliable and free from any econometrics problems. The results from Table 4 presented the diagnostic tests for all three models. The diagnostics checking performed in these models has passed four major tests of serial correlation, functional form, normality, and heteroscedasticity, given that the probability value for each test is larger than a 10% significant level. This means that the stochastic error term is white noise, the specifications of the models are well specified, normally distributed with zero mean and constant variance, hence the models are robust.

To further enhance the reliability of our results, the three models were diagnosed with stability tests using the cumulative sum of recursive residuals (CUSUM) and the cumulative sum of the square of recursive residuals (CUSUMSQ). The results displayed in Figure 2 suggest that the coefficients (showed by the blue line) of the three models are stable and consistent as the results are still within the critical bound (represent the two red lines). This implies that the obtained results from this research paper can be used for policy inference.

**Table 4.** Results of Diagnostic Checking.

| Model | A. Serial Correlation [*p*-Value] | B. Functional Form [*p*-Value] | C. Normality [*p*-Value] | D. Heteroscedasticity [*p*-Value] |
|---|---|---|---|---|
| Model of Economic Growth | 0.771 [0.475] | 1.944 [0.177] | 0.524 [0.769] | 1.053 [0.448] |
| Model of Income Distribution | 1.17 [0.31] | 0.79 [0.39] | 1.24 [0.53] | 0.71 [0.76] |
| Model of Environmental Quality | 2.77 [0.110] | 1.76 [0.22] | 1.54 [0.46] | 1.71 [0.16] |

Note. 1. The numbers in brackets [ ] are *p*-value. 2. The diagnostic test performed as follows A. Lagrange multiplier test for residual serial correlation; B. Ramsey's RESET test using the square of the fitted values; C. Based on a test of skewness and kurtosis of residuals; D. Based on the regression of squared fitted values.

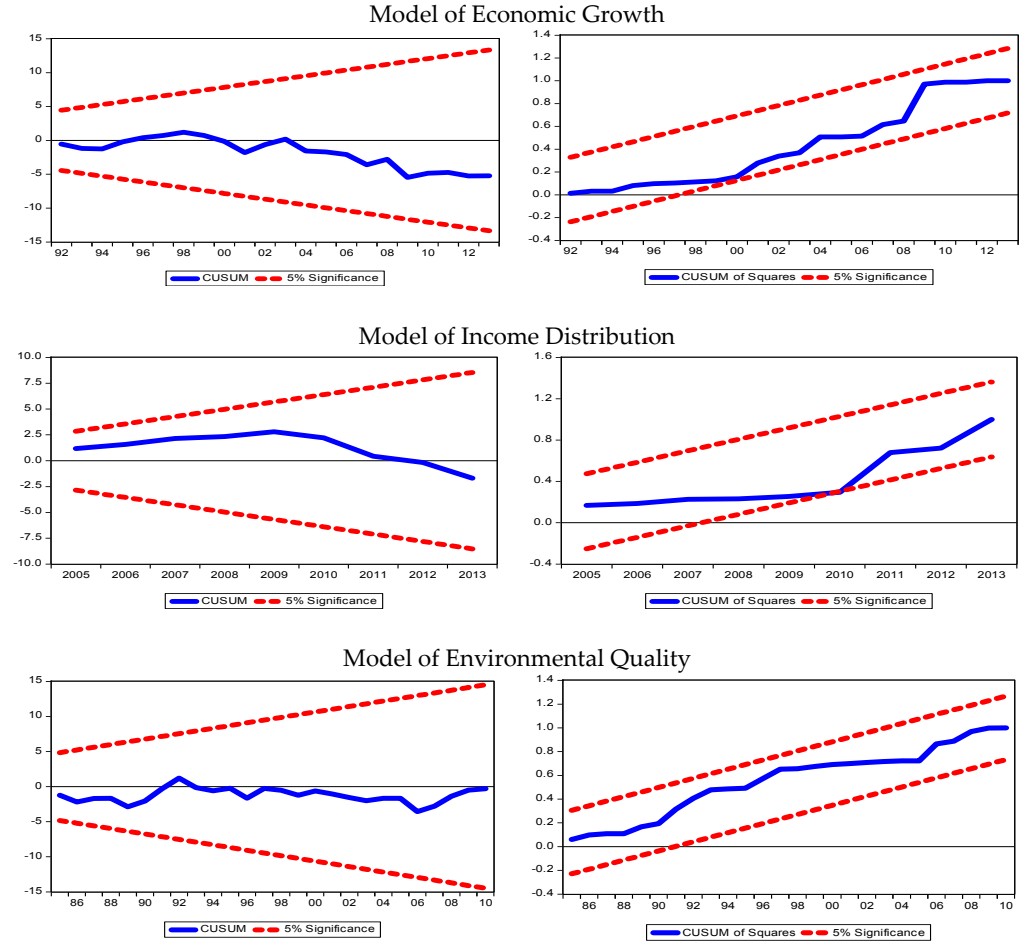

**Figure 2.** CUSUM (**left**) and CUSUMSQ (**right**).

After confirming the reliability of the proposed models, we now move to the core part of the analysis. Table 5 shows results for the long run elasticities for each model. Based on the output from the model of economic growth, it was found that FDI had a weaker positive impact on the Singapore economic growth at 10% significant level. Meanwhile, HC, TO and FD had stronger positive impact on the Singapore economy. An increase of FDI, HC, TO and FD by 1% could increase Singapore economic growth by 0.07%, 0.80%, 0.57% and 0.63%, respectively. The biggest impact on economic growth in Singapore is human capital development (HC). Realizing the importance of HC on generating growth,

over the years, public expenditure on education has consistently been the second highest (after defense) in the government's annual fiscal budget in Singapore. Next, the positive impact of FD on growth is in line with previous studies by Hermes and Lensink (2003), Durham (2004) and Alfaro et al. (2004) who find that countries with better financial systems and financial market regulations can exploit FDI more efficiently and achieve a higher growth rate. The positive relationship between TO and growth in Singapore may come from three channels as suggested by Anderson and Babula (2008). First, trade provides access to foreign intermediate inputs and technologies that are not available in Singapore as this country is a small city-state with no natural resources. Second, trade also increases market sizes for new product varieties. Third, trade allows for diffusion of general knowledge across geographical boundaries that further facilitate the R&D process and subsequent innovation. Among past empirical studies that support the positive openness-growth nexus include Wacziarg and Welch (2008), Squalli and Wilson (2011) and Sakyi et al. (2014). Next, the positive impacts of FDI on growth validate FDI-led growth hypothesis for Singapore and this outcome could be driven by the positive impact of absorptive capacity variables (HC, FD and TO) on growth. The availability of high skilled workers, stability in the financial market and the implementation of trade liberalization, increase the confidence of foreign investors to invest more in this country. LAB and DI, on the other hand, are found to have a negative impact on growth. However, given that DI is not significant at any level this means that DI is not suited to explain growth in Singapore.

On the second model, we found that the rise of DI and FD can reduce the GINI coefficient, thus improving income distribution in the country. Precisely, a 1% increase in DI and FD will reduce the GINI coefficient by 0.99% and 0.55%. The negative relationship between DI and GINI is explained by the intuition that increases in domestic investment spending means more people getting jobs, which implies that more people are earning, thereby putting a downward pressure on income inequality. Meanwhile, recent literature reviews have taken a stance that financial development is an important factor in reducing income inequality (Ang 2010; Clarke et al. 2006). According to Galor and Zeira (1993), easy access to financial resources boosts investment activities that directly increase the income of poor segments of the population by generating employment opportunities. Besides, easy access to financial resources enables poor people to feed their children and educate them for a better future that results in income distribution improvement in (Canavire-Bacarreza and Rioja 2009). FDI, on the other hand, has a positive relationship with GINI. This means that higher FDI inflows in Singapore will result to higher income disparity within the society when the multinational companies (MNC) is focusing on only recruiting high-skilled labor which in turn will result in an increase in income inequality (Lee and Vivarelli 2006). The last two determinants GDP and TO were not significant at any level and thus, failed to have any relationship with income distribution in Singapore.

Based on the model of environmental quality, it shows that both GDP and EN have a positive relationship with $CO_2$ emissions in Singapore. Ang (2007) and Halicioglu (2009) stated that higher economic development in the country might result in higher energy consumption and thus lead towards a higher release of $CO_2$ emissions. The expansion of the Singapore economy that led towards environmental degradation in this study were similar to the research findings by Ali et al. (2017) in Singapore. Meanwhile, the positive relationship detected between GDP and $CO_2$ emissions were in line with previous studies such as, Cole (2004). Next, negative signs for both FDI and TO on $CO_2$ emissions mean that higher FDI inflows and increase in trade liberalization have successfully reduced environmental degradation in the country. To be more precise, a 1% increases in FDI and TO will decrease the release of $CO_2$ emissions by 0.22% and 0.34%, respectively. FDI inflows received by Singapore might focus on modern and cleaner technologies in the production process to ensure best environmental practices that could reduce air pollution. Furthermore, the negative sign of TO shows that the country could specialize in clean and service intensive products for export and import pollution-intensive products from their trading partners. Lastly, FD does not influence $CO_2$ emissions in Singapore, given that there is no significant level detected between these two variables.

**Table 5.** Estimation of Long-Run Elasticities.

| Model of Economic Growth | | Model of Income Distribution | | Model of Environmental Quality | |
| --- | --- | --- | --- | --- | --- |
| **LNGDP** | | **LNGINI** | | **LNCO$_2$** | |
| **Variables** | **Coefficient** | **Variables** | **Coefficient** | **Variables** | **Coefficient** |
| LNLAB | −0.781 *** | LNGDP | −0.190 | LNGDP | 1.589 ** |
| LNDI | −0.052 | LNDI | −0.993 *** | LNFDI | −0.221 *** |
| LNFDI | 0.072 * | LNFDI | 0.198 *** | LNFD | 0.860 |
| LNHC | 0.804 *** | LNTO | 0.242 | LNEN | 0.172 *** |
| LNTO | 0.579 *** | LNFD | −0.548 ** | LNTO | −0.314 *** |
| LNFD | 0.634 *** | C | 14.533 *** | C | 6.256 *** |
| C | 2.602*** | | | | |

Note: 1. ***, **, * indicate significant at 1%, 5% and 10% significant level, respectively.

Table 6 shows the outcome for short run elasticities based on error correction model (ECM). Special attention is given to the expected results based on lag 0 on each model. Based on the model of economic growth, DI, FDI, and HC are found to have a positive relationship with growth in the short run. LAB, TO and FD do not have any relationship with growth as it is not significant relationship at any level. Next, based on the model of income distribution, it is revealed that GDP, DI and FDI have a significant relationship with income distribution, captured by Gini coefficient. Based on these three variables, GDP and DI exhibit negative sign while FDI exhibit positive sign. Lastly, based on the model of environmental quality, it is shown that FDI improve the environmental quality in the short run. To conclude, in the short run, it is found that FDI inflows have led to greater economic growth and environmental quality but at the same time increases the income disparity within the country. TO and FD did not influence any of the three models in the short run.

**Table 6.** Estimation of Short Run Restricted Error Correction Model (ECM).

| Model of Economic Growth | | Model of Income Distribution | | Model of Environmental Quality | |
| --- | --- | --- | --- | --- | --- |
| **Variables** | **Coefficient** | **Variables** | **Coefficient** | **Variables** | **Coefficient** |
| $\Delta$LNLAB$_t$ | 0.018 | $\Delta$LNGINI$_{t-1}$ | 0.198 * | $\Delta$LNCO$_{2t-1}$ | −0.259 |
| $\Delta$LNDI$_t$ | 0.150 * | $\Delta$LNGINI$_{t-2}$ | 0.310 ** | $\Delta$LNCO$_{2t-2}$ | −0.573 ** |
| $\Delta$LNFDI$_t$ | 0.019 * | $\Delta$LNGINI$_{t-3}$ | −0.204 * | $\Delta$LNGDP$_t$ | −0.239 |
| $\Delta$LNFDI$_{t-1}$ | 0.003 *** | $\Delta$LNGDP$_t$ | −0.202 * | $\Delta$LNGDP$_{t-1}$ | −0.012 |
| $\Delta$LNHC$_t$ | 0.181 * | $\Delta$LNGDP$_{t-1}$ | 0.189 | $\Delta$LNGDP$_{t-2}$ | −0.219 |
| $\Delta$LNTO$_t$ | 0.139 | $\Delta$LNGDP$_{t-2}$ | 0.152 | $\Delta$LNGDP$_{t-2}$ | −0.458 |
| $\Delta$LNTO$_{t-1}$ | −0.198 ** | $\Delta$LNGDP$_{t-3}$ | −0.326 ** | $\Delta$LNFDI | −0.274 * |
| $\Delta$LNFD$_t$ | −0.119 | $\Delta$LNDI$_t$ | −0.167 * | $\Delta$LNFDI$_{t-1}$ | 0.101 |
| $\Delta$LNFD$_{t-1}$ | −0.159 ** | $\Delta$LNDI$_{t-1}$ | −0.197 ** | $\Delta$LNFDI$_{t-2}$ | −0.077 |
| $\Delta$LNFD$_{t-2}$ | −0.124 * | $\Delta$LNDI$_{t-2}$ | 0.038 | $\Delta$LNFDI$_{t-3}$ | −0.277 * |
| ECT$_{t-1}$ | −0.674 *** | $\Delta$LNDI$_{t-3}$ | 0.236 *** | $\Delta$LNFD | −0.053 |
| | | $\Delta$LNFDI$_t$ | 0.022 ** | $\Delta$LNFD$_{t-1}$ | 0.557 |
| | | $\Delta$LNFDI$_{t-1}$ | −0.05 ** | $\Delta$LNFD$_{t-2}$ | −1.850 ** |
| | | $\Delta$LNFDI$_{t-2}$ | −0.024 ** | $\Delta$LNFD$_{t-3}$ | 1.561 ** |
| | | $\Delta$LNFDI$_{t-3}$ | 0.034 ** | $\Delta$LNENY$_t$ | −0.028 |
| | | $\Delta$LNTO$_t$ | −0.074 | $\Delta$LNENY$_{t-1}$ | −0.895 ** |
| | | $\Delta$LNTO$_{t-1}$ | −0.045 | $\Delta$LNENY$_{t-2}$ | −0.565 |
| | | $\Delta$LNTO$_{t-2}$ | 0.134 | $\Delta$LNTO$_t$ | −0.093 |
| | | $\Delta$LNTO$_{t-3}$ | −0.130 * | $\Delta$LNTO$_{t-1}$ | 0.471 |
| | | $\Delta$LNFD$_t$ | −0.030 | $\Delta$LNTO$_{t-2}$ | −0.229 |
| | | $\Delta$LNFD$_{t-1}$ | −0.121 ** | $\Delta$LNTO$_{t-3}$ | 2.478 ** |
| | | $\Delta$LNFD$_{t-2}$ | 0.183 *** | ECT$_{t-1}$ | −0.394 ** |
| | | $\Delta$LNFD$_{t-3}$ | −0.130 * | | |
| | | ECT$_{t-1}$ | −0.354 *** | | |
| *R* square | 0.99 | *R* square | 0.99 | *R* square | 0.98 |
| Ad.*R* square | 0.99 | Ad.*R* square | 0.99 | Ad.*R* square | 0.94 |

Note: 1. ***, **, * indicate significant at 1%, 5% and 10% significant level, respectively.

The long run elasticities on each model were supported by the negative and significant value of error correction term (ECT). ECT represents the speed of adjustment for each model and the negative value means that the variables will converge in the long run. The highest speed of adjustment is detected for a model of economic growth ($-0.67$), followed by a model of environmental quality ($-0.39$) and a model of income distribution ($-0.35$). Approximately, 67 per cent, 39 per cent and 35 per cent disequilibria from the previous year's shock reconverged on the long run equilibrium in the current year. Overall, the R-square for all three models suggests that almost 94 per cent and above of the variables in equations for Singapore explain the dependent variable.

The Toda and Yamamoto (1995) granger non-causality test through vector autoregressive (VAR) model was conducted to investigate the direction of causality between determinants for model economic growth, model of income distribution and model of environmental quality. This method is valid regardless of whether a series is I (0), I (1) or I (2), non-cointegrated or cointegrated of any arbitrary order. The optimum lag for each model was detected using VAR lag order selection based on AIC. The optimum lag detected for a model of economic growth was 4. Meanwhile, the optimum lag for a model of income distribution as well as a model of environmental quality was 5. One extra lag (dmax = 1) was added to the optimal lag of the VAR model for implementing the Granger non-causality test using the Toda and Yamamoto approach. Next, to make sure that each model was dynamically stable, inverse roots of AR polynomial was performed. Based on Figure 3 below, it can be confirmed that all models are dynamically stable given that the inverted roots (dotted blue) for each model are all strictly inside the circle.

The results of the Toda-Yamamoto granger non-causality test is shown in Table 7 below while the figure illustration can be viewed in Figure 4.

Based on the model of economic growth, we can confirm that four bidirectional causalities existed between (a) GDP and LAB, (b) LAB and FD, (c) LAB and TO and (d) LAB and HC. This relationship shows that Singapore depends highly on its labor to generate economic growth, deepening its financial sector, increases the production for export activities and advancement in the labor skills through human capital development. The unidirectional causality was detected between FD on GDP, TO and FDI. A strong financial sector in Singapore could assist the country to achieve a higher growth rate, increasing its value of trade as well as attracting a higher amount of FDI into the country. TO had a unidirectional causality with FDI and this supports Bhagwati's hypothesis that the more open the country, the more attractive the country will be a hub of investment by foreign investors. Next, HC unidirectional granger caused TO and GDP of the country. With high skilled labor available in the country, it will produce more output and a higher volume of output will increase exports for international trade. FDI had a unidirectional relationship with LAB, which means that through foreign investment, more MNCs will open their business in the country, thus giving an opportunity to the locals to get jobs. Lastly, under the model of growth, we found that DI unidirectional granger caused FDI, FD, and LAB.

Next, based on the model of income distribution, we found ten strong bidirectional causalities at 1% significant level, detected between (a) GINI and FD, (b) GINI and FDI, (c) GINI and DI, (d) FD and FDI, (e) FD and DI, (f) FD and GDP, (g) FDI and GDP, and (h) DI and GDP. Meanwhile, unidirectional causality was found from GINI to GDP. This means that the economic growth of the country is a prerequisite for better income distribution in the country. Next, unidirectional causality was found from TO to FD. The rapid expansion of Singapore's financial market leads to an introduction of new financial products and thus increases exports of the country. We also found that TO had a unidirectional causality to GDP, DI, and FDI. A unidirectional relationship running from FDI to DI was also shown in the model of income distribution.

Model of Economic Growth
Inverse Roots of AR Characteristic Polynomial

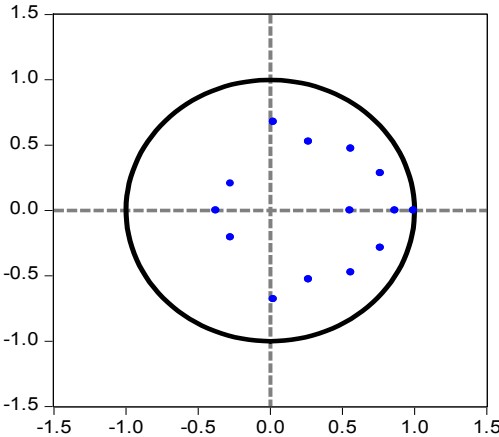

Model of Income Distribution
Inverse Roots of AR Characteristic Polynomial

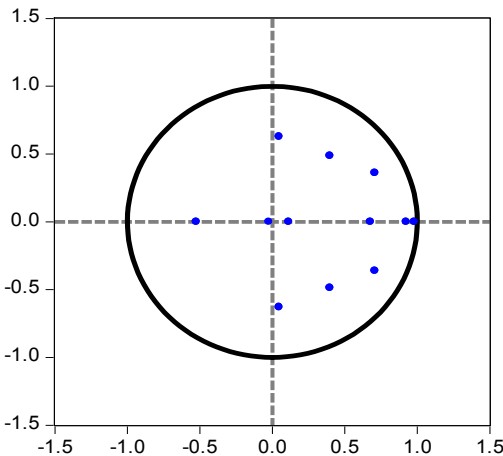

Model of Environmental Quality
Inverse Roots of AR Characteristic Polynomial

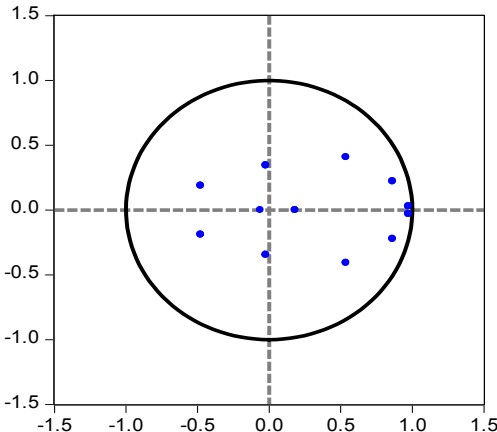

**Figure 3.** AR Roots Graph.

**Table 7.** Toda Yamamoto Granger Non-Causality Test.

| Dependent Variable | Direction of Causality | | | | | | |
|---|---|---|---|---|---|---|---|
| | Model of Economic Growth | | | | | | |
| | Short Run | | | | | | |
| Singapore | $LNGDP_t$ | $LNLAB_t$ | $LNDI_t$ | $LNFDI_t$ | $LNHC_t$ | $LNTO_t$ | $LNFD_t$ |
| $LNGDP_t$ | - | 19.938 *** [0.005] | 3.881 [0.422] | 1.774 [0.777] | 20.673 *** [0.000] | 19.745 *** [0.000] | 10.500 ** [0.032] |
| $LNLAB_t$ | 191.75 *** [0.000] | - | 57.922 *** [0.000] | 128.437 *** [0.000] | 24.035 *** [0.000] | 71.408 *** [0.000] | 45.608 *** [0.000] |
| $LNDI_t$ | 3.894 [0.420] | 0.402 [0.982] | - | 5.956 [0.202] | 2.112 [0.715] | 3.792 [0.434] | 3.395 [0.494] |
| $LNFDI_t$ | 5.622 [0.229] | 7.313 [0.120] | 18.129 *** [0.001] | - | 5.065 [0.280] | 11.991 ** [0.017] | 9.747 ** [0.044] |
| $LNHC_t$ | 7.369 [0.117] | 14.154 *** [0.006] | 2.230 [0.693] | 3.078 [0.544] | - | 6.555 [0.161] | 4.284 [0.368] |
| $LNTO_t$ | 6.261 [0.184] | 9.401 * [0.051] | 6.619 [0.157] | 5.999 [0.199] | 12.345 ** [0.015] | - | 7.957 * [0.093] |
| $LNFD_t$ | 4.133 [0.388] | 7.852 * [0.097] | 8.136 * [0.086] | 0.207 [0.995] | 7.541 [0.109] | 7.255 [0.123] | - |
| | **Model of Income Distribution** | | | | | | |
| | $LNGINI_t$ | $LNGDP_t$ | $LNDI_t$ | $LNFDI_t$ | $LNTO_t$ | $LNFD_t$ | |
| $LNGINI_t$ | - | 6.355 [0.273] | 11.767 ** [0.038] | 10.468 * [0.063] | 5.719 [0.334] | 24.030 *** [0.000] | |
| $LNGDP_t$ | 39.788 *** [0.000] | - | 38.305 *** [0.000] | 27.970 *** [0.000] | 27.671 *** [0.000] | 33.739 *** [0.000] | |
| $LNDI_t$ | 86.420 *** [0.000] | 31.847 *** [0.000] | - | 86.737 *** [0.000] | 113.112 *** [0.000] | 86.881 *** [0.000] | |
| $LNFDI_t$ | 35.622 *** [0.000] | 193.346 *** [0.000] | 6.939 [0.225] | - | 51.928 *** [0.000] | 207.302 *** [0.000] | |
| $LNTO_t$ | 0.643 [0.985] | 0.288 [0.997] | 0.728 [0.981] | 0.600 [0.988] | - | 0.650 [0.985] | |
| $LNFD_t$ | 56.103 *** [0.000] | 49.001 *** [0.000] | 81.610 *** [0.000] | 48.775 *** [0.000] | 32.253 *** [0.000] | - | |
| | **Model of Environmental Quality** | | | | | | |
| | $LNCO_{2t}$ | $LNGDP_t$ | $LNFDI_t$ | $LNFD_t$ | $LNEN_t$ | $LNTO_t$ | |
| $LNCO_{2t}$ | - | 4.769 [0.444] | 5.285 [0.382] | 2.868 [0.720] | 2.584 [0.763] | 2.622 [0.758] | |
| $LNGDP_t$ | 4.026 [0.545] | - | 3.499 [0.623] | 3.045 [0.692] | 2.766 [0.735] | 2.667 [0.751] | |
| $LNFDI_t$ | 53.108 *** [0.000] | 21.864 *** [0.000] | - | 77.163 *** [0.000] | 31.691 *** [0.000] | 26.350 *** [0.000] | |
| $LNFD_t$ | 3.326 [0.649] | 4.048 [0.542] | 4.651 [0.459] | - | 3.513 [0.621] | 6.700 [0.243] | |
| $LNEN_t$ | 30.419 *** [0.000] | 46.416 *** [0.000] | 49.076 *** [0.000] | 29.603 *** [0.000] | - | 47.636 *** [0.000] | |
| $LNTO_t$ | 63.906 *** [0.000] | 174.297 *** [0.000] | 108.501 *** [0.000] | 386.316 *** [0.000] | 149.461 *** [0.000] | - | |

Note: 1. ***, **, * indicate significant at 10%, 5% and 1% significant level, respectively. The optimum lag detected was 4 based on AIC and K + dmax = 5 for a model of economic growth. For a model of income distribution and model of environmental quality, the number of optimum lag detected was 5 based on AIC, and K + dmax = 6.

Based on the model of environmental quality, only three bidirectional causalities were found running between (a) TO and FDI, (b) EN and FDI and (c) EN and TO. The unidirectional causality

running from CO$_2$ emissions to TO, EN and FDI reflected that the country's environmental condition could influence openness to trade, level of energy consumption and level of FDI inflows. Lastly, the unidirectional causality can be seen from GDP to TO, EN and FDI. This means that per income capita growth of the country can be a prerequisite for the practice of trade liberalization, stimulation of higher energy used, as well as reasons for higher foreign investment. Similar to GDP, the unidirectional causality was also found running from FD to EN, TO and FDI. The illustration of granger causality can be seen in Figure 4 below:

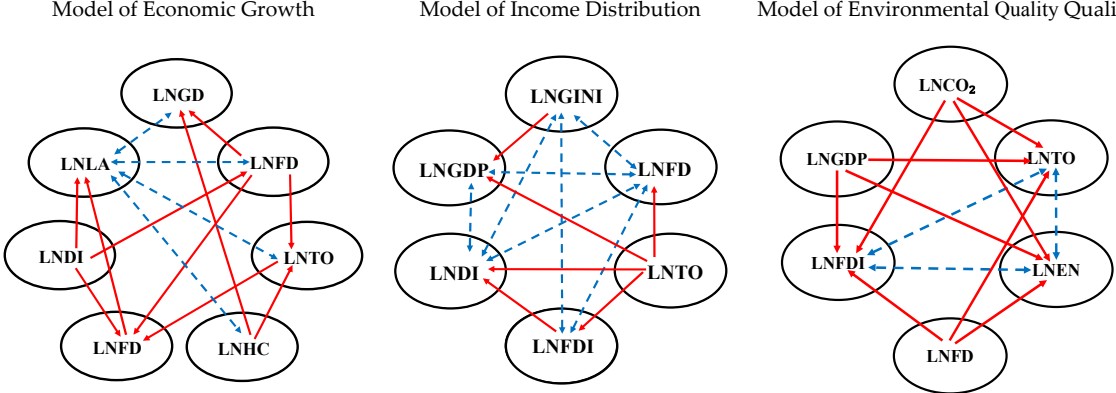

**Figure 4.** Toda-Yamamoto granger non-causality test. Note: The red line refers to unidirectional causality while the blue dotted line (feedback hypothesis) refers to bi-directional causality.

## 6. Conclusions

This study focuses on assessing the impact of FDI inflows on economic growth, income distribution and environmental quality in Singapore. To summarize, the outcome of the analysis showed that FDI inflows could only bring favorable impact on economic growth and environmental quality. Based on the income distribution model, higher FDI inflows could worsen the income equality in the country. Thus, based on this case, the government can play its role to provide more incentives to foreign companies that are able to provide more job opportunities not only to high-skilled workers but also to semi-skilled workers. Besides FDI, it also found that TO and FD could also play their part as drivers for SD. Singapore needs to remain open, not only to international investment flows but also to their international trade activities as well as strengthening their financial market development in order to create a competitive environment that could lead towards the realization of sustainable development. Lastly, the output from causality test could shed more ideas on future policy recommendations.

**Author Contributions:** All authors contributed equally to this work.

**Conflicts of Interest:** The authors declare no conflicts of interest.

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
