# Peer review of "Does Foreign Direct Investment Successfully Lead to Sustainable Development in Singapore?"

_economies, doi:10.3390/economies5030029_

Reviewer 1 Report

Overall, this is a well-written paper with a focus on interesting topics. However, there are few limitations in which the author should address to improve the quality of this paper.

They are:

a) The author may want to explain the theoretical foundation of control variables in equations 1, 3 and 5.

b) Why FD, M2 over GDP, is a good proxy as financial development, given that other proxies have been used also?

c) Why CO2 is a good proxy for environmental quality? Others have been used in similar study.

d) Should the author consider environmental Kuznet curve in equation 5?

e) Refer to Table 5 in page 11. Why the number of lags are not consistent with the lags presented in Table 2?

f) If ECM can be formulated, why should the author use Toda and Yamamoto approach to study Granger causality?

Author Response

Dear reviewer 1.

Thank you very much for constructive comments. We have try our best to revise our paper based on your list of comments. 

Regards

Reviewer 1

Overall, this is a well-written paper with a focus on interesting topics. However, there are few limitations in which the author should address to improve the quality of this paper.

a) The author may want to explain the theoretical foundation of control variables in equations 1, 3 and 5.

Brief introduction of the original model is added for clarification.

b) Why FD, M2 over GDP, is a good proxy as financial development, given that other proxies have been used also?

The used of M2 over GDP in our study because it is the most popular proxy used to capture financial development such as domestic credit to private sector. The data is more complete as compared to other proxies.

c) Why CO2 is a good proxy for environmental quality? Others have been used in similar study.

We have added brief reasons why we used CO2 emissions.

d) Should the author consider environmental Kuznet curve in equation 5?

The real GDP square (which testify EKC) was removed in the model was not something new. It has been applied by many other researchers such as Hossain (2011), Al Mulali and Tang (2013), and Shahbaz et al. (2013). For this research paper, we would like to have a consistent model for all three pillars of SD. If we introduced EKC for model of environmental quality, we also need to introduced GDP square for model of income distribution to capture Kuznets curve. However, we prefer to run without it.

e) Refer to Table 5 in page 11. Why the number of lags are not consistent with the lags presented in Table 2?

Missing information. Correction done.

f) If ECM can be formulated, why should the author use Toda and Yamamoto approach to study Granger causality?

Given that there is a mix stationarity for the variables, thus, to test causality, we need to perform Toda Yamamoto Granger causality test. If all the variable are only stationary at first difference, then we can proceed with VECM granger causality test.

Reviewer 2 Report

1. The objective of the paper is not clearly stated.
2. What did the author mean with 'an alternative mechanism' in line 64?
3. The author should mention, may be in couple of line, how the models in line 143, 160 and 165 are derived to help readers having a clear understanding.
4. Be careful with the consistency of using the symbol of error term.  In the model (in line 160), error term is represented by v but in the sentence (line 161), u is used.  The same problem for error terms in line 186 and 204.
5. The bound of order (p q r s t u v) in line 163 should be separated by comma (p,q,r,s,t,u,v).  The same problem in line 187 and line 205.
6. The letter 't' should be subscript in the model (line 143, line 165, and line 189)
7.  I recommend the author should not use the same symbol for coefficient in all models (line 160, line 185 and line 203) to avoid confusion.
8. Please put bracket Equation (2), Equation (4) and Equation (6) in line 207.
9. Please change the symbol of lambda for hypothesis in line 207 and line 208 if comment 7 is applied.
10.In line 298, the author should write 0.55% ( two decimal point)
11. What does it mean "the variables in each variable" in line 341.

Author Response

Dear reviewer 2.

Thank you very much for constructive comments. We have try our best to revise our paper based on your list of comments. 

Regards

1. The objective of the paper is not clearly stated.

Objective of the research paper is refined.

2. What did the author mean with 'an alternative mechanism' in line 64?

Wrong sentences. Correction done.

3. The author should mention, may be in couple of line, how the models in line 143, 160 and 165 are derived to help readers having a clear understanding.

The origin of the model is added.

4. Be careful with the consistency of using the symbol of error term. In the model (in line 160), error term is represented by v but in the sentence (line 161), u is used. The same problem for error terms in line 186 and 204.

Replace v and u to μ for all three equations

5. The bound of order (p q r s t u v) in line 163 should be separated by comma (p,q,r,s,t,u,v). The same problem in line 187 and line 205.

Correction done

6. The letter 't' should be subscript in the model (line 143, line 165, and line 189)

Correction done

7. I recommend the author should not use the same symbol for coefficient in all models (line 160, line 185 and line 203) to avoid confusion.

Correction done

8. Please put bracket Equation (2), Equation (4) and Equation (6) in line 207.

Correction done

9. Please change the symbol of lambda for hypothesis in line 207 and line 208 if comment 7 is applied.

Correction done

10.In line 298, the author should write 0.55% ( two decimal point)

Correction done

11. What does it mean "the variables in each variable" in line 341.

Wrong sentences. Correction done.

Reviewer 3 Report

line 191-192, correction for GDP EN....should be a comma after GDP, and what is kg of oil equivalent as it is not clear

2.missing references for: a. Ang (2010)

                                        b.Dick (2010)

                                        c.Feenstra (1997)

3.why is references numbered from 1-116 with references not aligned, 

4.why was the variance inflation  factor not shown/conducted as it could explain the high r squared as there could  possibly be the presence of 
multi collinearity

5.please specify the specific test for serial correlation,functional form, normality and heteroscedasticity , eg for serial correlation, is it Breusch-Godfrey? 

6. For table Table 5 with regard to the Notes....missing word is 'respectively'?

Author Response

Dear reviewer 3.

Thank you very much for constructive comments. We have try our best to revise our paper based on your list of comments. 

Regards

Reviewer 3

1. line 191-192, correction for GDP EN....should be a comma after GDP, and what is kg of oil equivalent as it is not clear

Correction done.

2.missing references for:

a. Ang (2010)

Ang J. B. (2010). Determinants of private investment in Malaysia: What causes the post- crisis slumps? Contemporary Economic Policy, 28: 378–391.

b. Dick (2010)

Dick, C. (2010). Do environmental characteristics influence foreign direct investment growth? Across national study,1990–2000. International Journal of Comparative Sociology, 51: 192–210.

c. Feenstra and Hanson (1997)

Feenstra, R. C., & Hanson, G. H. (1997). Foreign direct investment and relative wages: Evidence from Mexico’s maquiladoras. Journal of International Economics, 42: 371–393.

3.why is references numbered from 1-116 with references not aligned,

Technical error. Formatting done.

4.Why was the variance inflation factor not shown/conducted as it could explain the high r squared as there could possibly be the presence of multi collinearity

For ARDL estimation, we have run all other diagnostic test procedures such as serial correlation, functional form, normality, heteroscedasticity, , CUSUM and CUSUMSQ test. As we passed all the condition for these test, the analysis provide reliable results.

5.please specify the specific test for serial correlation functional form, normality and heteroscedasticity, eg for serial correlation, is it Breusch-Godfrey?

List of tests involved is added.

6. For table Table 5 with regard to the Notes....missing word is 'respectively'?

Add missing word, respectively

Round  2

Reviewer 4 Report

1.Descriptive Analysis on the variables used in the study such as min,max,mean, standard deviation to describe basic features/characteristics of the data which can provide the reader more understanding on the embedded information in the data.

Author Response

Dear reviewer

I have insert the descriptive table in the earlier stage of analysis to give more information of the data/variables to the readers.

Thank you for your valuable comment